# Effect of Aprotinin on Liver Injury after Transplantation of Extended Criteria Donor Grafts in Humans: A Retrospective Propensity Score Matched Cohort Analysis

**DOI:** 10.3390/jcm10225232

**Published:** 2021-11-10

**Authors:** Anna B. Roehl, Anne Andert, Karsten Junge, Ulf P. Neumann, Marc Hein, Felix Kork

**Affiliations:** 1Department of Anesthesiology, Faculty of Medicine, RWTH Aachen University, 52074 Aachen, Germany; mhein@ukaachen.de (M.H.); fkork@ukaachen.de (F.K.); 2Department of General, Visceral and Transplantation Surgery, Faculty of Medicine, RWTH Aachen University, 52074 Aachen, Germany; aandert@ukaachen.de (A.A.); uneumann@ukaachen.de (U.P.N.); 3Department of General and Visceral Surgery, Rhein-Maas Hospital, 52146 Würselen, Germany; karsten.junge@rheinmaasklinikum.de

**Keywords:** aprotinin, liver transplantation, human, extended donor criteria

## Abstract

The number of patients awaiting liver transplantation still widely exceeds the number of donated organs available. Patients receiving extended criteria donor (ECD) organs are especially prone to an aggravated ischemia reperfusion syndrome during liver transplantation leading to massive hemodynamic stress and possible impairment in organ function. Previous studies have demonstrated aprotinin to ameliorate reperfusion injury and early graft survival. In this single center retrospective analysis of 84 propensity score matched patients out of 274 liver transplantation patients between 2010 and 2014 (OLT), we describe the association of aprotinin with postreperfusion syndrome (PRS), early allograft dysfunction (EAD: INR 1,6, AST/ALT > 2000 within 7–10 days) and recipient survival. The incidence of PRS (52.4% vs. 47.6%) and 30-day mortality did not differ (4.8 vs. 0%; *p* = 0.152) but patients treated with aprotinin suffered more often from EAD (64.3% vs. 40.5%, *p* = 0.029) compared to controls. Acceptable or poor (OR = 3.3, *p* = 0.035; OR = 9.5, *p* = 0.003) organ quality were independent predictors of EAD. Our data do not support the notion that aprotinin prevents nor attenuates PRS, EAD or mortality.

## 1. Introduction

Increasing organ shortage for liver transplantation is a major challenge for transplant hepatology. To address this situation and to overcome the discrepancy between organ demand and supply, marginal donors or extended criteria donors (ECD) are more often accepted to increase the number of available donor organs [1,2].

There is no standard definition of extended criteria donors. Eurotransplant considers liver graft donors as extended criteria donors if one of the following criteria is fulfilled: Donor age > 65 years, ICU stay with ventilation > 7 days, body mass index > 30, steatotic liver > 40% serum sodium > 165 mmol/L, SGPT > 105 U/L, SGOT > 90 U/L or serum bilirubine > 3 mg/dL (www.eurotransplant.org/organs/liver/, accessed on 22 September 2021). In the Eurotransplant region, available donor organs are primarily offered to patients matching the ABO blood group with the highest model of end stage liver disease (MELD) score nearest to the explantation site [3]. Organs form extended criteria donors, on the other hand, are offered to several centers closest to the explantation site. Remarkably, patients with a special urgency are exempted from the MELD-based allocation (e.g., acute liver failure, primary organ nonfunction). This leads to an increased acceptancy of organs from extended criteria donors being transplanted to increasingly sick patients [4,5].

Moreover, these liver grafts from extended criteria donors are especially susceptible to ischemia-reperfusion injury [6]. During the liver transplantation itself, one of the most crucial time points for the recipient is the reperfusion of the liver graft, potentially resulting in post-reperfusion syndrome (PRS) [7]. PRS has been shown to be associated with poorer short- and long-term outcomes, in particular hyperfibrinolysis, early allograft dysfunction (EAD) and mortality.

Several measures improving ischemia-reperfusion injury of the liver transplantation graft, and therefore attenuating PRS, have been subject to investigation and extensive discussion: changes in preservation solution, external organ perfusion as well as pharmacologic pre-conditioning by, among others, aprotinin [6]. Aprotinin is a reversible binding, competitive serine protease inhibitor widely used to reduce blood loss during heart as well as liver surgery [8,9,10,11,12,13]. One of the first descriptions of the applications of aprotinin in liver transplant patients has been described by Neuhaus and colleagues [11,12,14]. Aprotinin activates plasminogen and has been demonstrated to prevent microthrombosis, to improve microcirculation and consequently oxygen supply, and moreover, to ameliorate systemic inflammatory response [9,15,16]. Aprotinin reduces liver ischemia reperfusion injury in animal models [17,18] and improves 1-month graft survival in liver transplant recipients [9] The double-blinded European Multicenter Study on the Use of Aprotinin in Liver Transplantation (EMSALT) found that aprotinin reduced hyperfibrinolysis and consequently led to a 50% reduction of blood loss and a 30% reduction of transfusion requirement [10]. However, aprotinin was temporarily suspended as preliminary results from the BART trial demonstrated higher mortality for patients receiving aprotinin [19].

However, for the above mentioned reasons, aprotinin was an essential element in the perioperative management when transplanting liver grafts from extended criteria donors at our center. We therefore conducted a single-center retrospective analysis to investigate the effect of aprotinin on intraoperative and postoperative outcomes of liver graft recipients compared to a cohort of propensity score matched patients who were not treated with aprotinin.

## 2. Materials and Methods

### 2.1. Patients

The local ethics committee (University Hospital Aachen, EK 291/13) approved the analysis and waived the requirement of informed consent. From January 2010 to June 2015, we performed 274 liver transplantations in our tertiary care university hospital. Records of patients who received a liver transplantation during the observation period from 2010 to 2014 were reviewed for the intraoperative treatment of aprotinin. Patients were then matched by organ quality, donor age and CIT (for detailed description please see 2.8 Statistics).

### 2.2. Donor Data

The covering letter from Eurotransplant provided the donor data: age, sex, body mass index, cold ischemic time (CIT) as well as sodium concentration, alanine transaminase (ALT), aspartate transaminase (AST) and bilirubin (Bili).

### 2.3. Donor Organ Assessment

The backtable surgeon assessed the liver graft macroscopically by inspection and palpations as described before [20], assessing liver texture, yellowness, absence of scratch marks and round edges [21]. The organs were classified as either good, acceptable or poor quality following the Eurotransplant criteria. Macro- and microvesicular fat contents were determined by histology and were described as affected hepatocytes in percentage [7] including microsteatosis (MIS; the cytoplasm of the hepatocyte contains multiple tiny lipid vesicles without nuclear dislocation) or macrovesicular steatosis (the cytoplasm of the hepatocyte contains a univacuole lipid vesicle with nuclear displacement) [22].

### 2.4. Liver Transplantation Management

Liver transplantation was performed using an extracorporeal venovenous/portalvenous bypass. Bypass, surgical and anesthesiologic management as well as a peri- and postoperative immune suppression regimen have been described earlier in detail [7,20]. The patients received a maximum of one liter of balanced electrolyte solution. Adjacent volume replacement was held up by the transfusion of fresh frozen plasma (FFP) in order to anticipate coagulation disorders. Transfusion triggers for red blood cell units (RBC) were dependent on the patient’s comorbidities and transfusions were conducted at the discretion of the providing anesthesiologist. Our standard operation procedure (SOP) scheduled a thrombelastometry (TEM, Rotem Delta^®^,Werfen, Muenchen Germany) after the induction of anesthesia as well as 15–30 min and 45–60 min after reperfusion for the early correction of coagulation disorders [23].

### 2.5. Aprotinin Application

After allocation of the organ, the treating anesthesiologist and transplant surgeon decided jointly considering the following three aspects: (i) the visual assessment of the liver; (ii) to the CIT; and (iii) the donor age, whether the patient should receive aprotinin in order to attenuate PRS and early allograft dysfunction. This was a shared clinical decision without a predefined algorithm. Aprotinin infusion was started immediately after the surgical incision with a testing dose of 1 mL (equivalent to 10,000 IE) to rule out any allergic reaction. After that, aprotinin was infused at a rate of 2 × 10^6^ IE/h, a rate of 4 × 10^6^ IE/h during the an-hepatic phase and was reduced to 2 × 10^6^ IE/h until the end of surgery.

### 2.6. Recipient Data

Recipient data were abstracted from the patient’s medical chart: recipient age, diagnosis leading to transplantation and the MELD score (MELD: 10 × (3.8 × ln(bilirubin [mg/dl]) + 11.2 × log_e_(INR) + 9.6 × ln(creatinine level [mg/dL]) + 6.4 × (etiology: 0 if cholestatic or alcoholic, 1 otherwise) [24] were recorded at the evaluation procedure before patients were enlisted for transplantation. Clinical chemistry data (creatinine, AST, ALT, Bili, GGT, GLDH) were extracted from the electronical chart after admission closest to the beginning of the surgery (preoperatively), at ICU admission immediately after surgery (postoperatively) and on days 1, 3, 7 and 14. The number of intraoperatively transfused units of red blood cell units (RBC; units), fresh frozen plasma (FFP), platelets, fibrinogen and 4 factor prothrombin complex concentrate (4F-PCC) were extracted from the paper-based anesthesia protocol. Hyperfibrinolysis was diagnosed by the intraoperatively conducted rotational thrombelastometry [25], Bilirubin, INR, AST/ALT, acute rejection (clinical diagnosis), surgical revisions, re-transplantation, sepsis, acute kidney injury (as defined by the KDIGO clinical practice guidelines; www.kdigo.org, accessed on 22 September 2021), need for renal replacement therapy (RRT), intensive care unit (ICU) length of stay (LOS). Early allograft dysfunction is defined as bilirubin ≥ 10mg/dL on postoperative day (POD) 7 and/or INR ≥ 1.6 on POD 7 and/or AST or ALT > 2000 IU/L within the first 7 days, were abstracted from the patients’ chart after the transplantation.

### 2.7. Postreperfusion Syndrome 

Postreperfusion syndrome was defined as the occurrence of one of the following criteria: (1) decrease in mean arterial pressure (MAP) of at least 30% at time of reperfusion; (2) administration of an intravenous bolus of norepinephrine >2 μg kg body weight (BW)^−1^; (3) increase of continuous norepinephrine (NE) infusion of ≥0.1 μg kg BW^−1^ within 5 to 30 min after reperfusion; or (4) initiation of continuous vasopressin infusion after reperfusion. According to our department’s SOP, PRS was treated as follows: (i) 0.5 mg atropine before reperfusion if heart rate < 80; (ii) NE boli and NE infusion to maintain MAP; (iii) epinephrine boli and infusion in the case of significant bradycardia with hypotension and the decrease of SVO2 during reperfusion; (iv) infusion of vasopressin if high doses of NA are necessary or NA therapy is ineffective.

### 2.8. Statistics

Using organ quality, donor age and CIT, we identified a propensity score matched control group with the nearest neighbor method (SPSS 24.0, MatchIt package for R, IBM Corporation, Armonk, NY, USA). Matching variables were selected a priori, as these were the criteria used to apply aprotinin at our center at that time. Differences between groups were analyzed using the t-test for continuous, Chi-Square test for categorical and variance analysis for repeated measurements for continuous variables over time. Kaplan–Meyers curves were generated to display the effect of aprotinin on patient and graft survival. A logistic regression analysis was calculated to estimate the effect of aprotinin on early allograft dysfunction and a linear regression to estimate the effect on peak AST. Covariates (recipient age, recipient sex, recipient BMI, donor age, donor BMI, donor AST, CIT and organ quality) were introduced because they were deemed relevant by the literature or by clinical judgement. Parameters were included in the multivariate approach if the univariate analysis was significant (SPSS 24.0). Results were displayed as mean and standard deviation or absolute and relative number of cases. Figures were created using Prism 6.0 (GraphPad Software, San Diego, CA, USA). A two-sided *p*-value ≤0.05 was considered statistically significant.

## 3. Results

### 3.1. Patients

A total of 274 patients received a liver transplantation graft during the five-year study period and 42 of these patients were treated with aprotinin. Whether the patient was treated with aprotinin to attenuate PRS and EAD had been a shared decision of the attending anesthesiologist and the transplant surgeon based on the following criteria: (i) organ quality (good, acceptable, poor); (ii) cold ischemia time; and (iii) age of the donor. In a propensity score analysis using these three criteria, 42 statistically similar patients were identified who were not treated with aprotinin and were assigned to the control group. Matching reduced the relative multivariate imbalance L1 (0.634 vs. 0.782) and the χ^2^ balance test showed no significant imbalance (χ^2^ = 1.08, df = 4, *p* = 0.897) in the matched cohort (*n* = 84).

### 3.2. Liver Graft Recipient Characteristics

Demographic and clinical characteristics of the study population are described in detail in Table 1. Patients who were treated with aprotinin intraoperatively during a liver graft transplantation were of similar age and sex and had a similar BMI compared to controls. Patients treated with aprotinin did not differ regarding the reason for liver graft transplantation, preoperative clinical chemistry, number of intraoperative transfusions and intraoperative complications. Notably, patients who were treated with aprotinin did tend to have a higher preoperative MELD score (19.9 ± 8.5 vs. 16.4 ± 8.6; *p* = 0.061).

### 3.3. Liver Graft Donor Characteristics

Demographic and clinical characteristics of the liver graft donors for the matched study population of liver graft recipients are described in Table 2. Donors did not differ in age, sex and ICU length of stay for recipients treated with aprotinin compared to controls. Overall organ quality, graft fat content and cold and warm ischemia time did not differ as well. However, donors for recipients treated with aprotinin had a higher BMI (34.7 ± 8.9 vs. 30.3 ± 7.5, *p* = 0.018) and a higher ALT (85.7 ± 114.2 vs. 43.6 ± 38.6, *p* = 0.032).

### 3.4. Outcomes

Differences in outcome between liver graft recipients who were treated with aprotinin compared to recipients who were not treated with aprotinin are displayed in Table 3. During the intraoperative period, recipients did not differ regarding the incidence of postreperfusion syndrome (52.4% vs. 47.6%; *p* = 0.414) and hyperfibrinolysis (7.1% vs. 9.5%; *p* = 1.0). Similarly, the number of transfusion units of packed red blood cells, fresh frozen plasma and platelets, as well as the amount of fibrinogen and four-factor prothrombin complex concentrate, did not differ between liver graft recipients who were treated with aprotinin compared to liver graft recipients who were not.

Liver graft recipients who were treated with aprotinin had more postoperative complications than recipients who were not treated with aprotinin—they suffered more often from early allograft dysfunction (64.3% vs. 40.5%; *p* = 0.029), suffered more often from acute kidney injury (48.8% vs. 26.2%; *p* = 0.033) and more often required renal replacement therapy (24.4% vs. 7.1%; *p* = 0.015) than controls. Multivariable regression analysis confirmed this finding: Intraoperative treatment with aprotinin was associated with a 4-fold risk (OR 4.12, 95%CI 1.21–14.00; *p* = 0.023), acceptable donor organ quality with a 5-fold risk (OR 4.95, 95%CI 1.26–19.5; *p* = 0.022) and poor donor organ quality with a 12-fold risk (OR 11.77, 95%CI 2.00–69.5, *p* = 0.007) of developing early allograft dysfunction. The multivariable regression analysis adjusted for recipient age, sex, BMI and MELD, donor age, BMI and AST, cold ischemic time and organ quality (Table 4). An additional sensitivity analysis reconfirmed this: Matching of a cohort using a propensity score including age, sex, BMI, MELD, donor age, donor BMI, donor AST, cold ischemia time and organ quality led to a cohort of 70 liver graft recipients, 35 of whom were treated with aprotinin and 35 of whom were not. In this alternate matched cohort, recipient and donor data did not differ in any of the above reported variables, but recipients still suffered more often from early allograft dysfunction when treated with aprotinin compared to controls (68.6% vs. 37.1%, *p* = 0.017; data not shown).

In accordance with the higher incidence of early allograft dysfunction, recipients who were treated with aprotinin had higher maximum AST (3193.6 ± 2273.5 vs. 1984.2 ± 1883.9; *p* = 0.010) and ALT (1611.3 ± 1152.4 vs. 975.2 ± 937.4; *p* = 0.007) after transplantation of the graft. A multivariable linear regression model confirmed this by demonstrating that having been treated intraoperatively with aprotinin was associated with a 1.324 U/L (95%CI 354–2.295) higher maximum AST after adjusting for recipient age, sex, BMI and MELD, donor age, BMI and AST, cold ischemic time and organ quality (Table 4).

Liver graft recipients treated with aprotinin did not differ regarding mortality from recipients not treated with aprotinin: 30-day mortality, 1-year mortality, and overall mortality (*p* > 0.152; Table 3) as well as recipient survival time and graft survival time did not differ from controls (Figure 1).

## 4. Discussion

In this single center retrospective analysis of 84 propensity score matched liver graft recipients of organs from extended criteria donors, we sought to determine the association of intraoperative treatment with aprotinin with hyperfibrinolysis, PRS, EAD and mortality. We found that patients receiving intraoperative aprotinin did not differ regarding the incidence of intraoperative hyperfibrinolysis, PRS or mortality compared to controls. However, liver graft recipients suffered more often from EAD (64% vs. 41%) and had higher postoperative peak transaminases compared to controls. In fact, multivariable regression analyses determined intraoperative treatment with aprotinin to be independently associated with a four-fold risk of EAD and an on average 1.300 U/L higher peak AST after liver graft transplantation in patients receiving organs from extended criteria donors.

All of the 84 liver graft recipients in this analysis were treated at the same center with the same operative technique (intraoperative venovenous/portalvenous bypass) [20] and with an SOP guided intra-operative management. Although this has led to a homogenous single center study sample, it also limits the external validity of the results. Limited external validity is a common problem when analyzing liver transplant patient data. In 2019, in Germany alone, there were 22 liver transplanting centers transplanting 1571 liver grafts, each with its own characteristic treatment modalities. However, this limitation will only be overcome by an efficient multicenter registry that collects comprehensive perioperative datasets of adequate granularity. Furthermore, the retrospective design of our analyses could have impaired data quality.

In our sample, the incidence of EAD was 52%. While the incidence of EAD has been reported in the literature between 23% [26] and 39% [27], the notably high incidence in our sample is most likely due to the fact that over 90% of the patients analyzed in this study received a liver graft from a donor that fulfilled at least one criterium as extended donor; that is, the high incidence of EAD may be attributable to the transplantation of marginal organs with a high proportion of moderate and poor organs containing above-average mircovesicular and macrovesicular fat. Evidence describing risk factors for EAD is rather scarce and moreover limited by the multitude of treatment modalities as described above. Evidence also suggests that mainly graft steatosis, recipient MELD score, CIT [28] and also donor BMI [26,27,29] are risk factors for EAD. Our sample may have been biased with regard to the fact, that donors for patients treated with aprotinin had a tendentiously higher BMI compared to controls. This may be due to fact that we considered only donor age, organ quality and CIT for matching, as those were the criteria for the clinical consensus decision to treat the graft recipient with aprotinin. Interestingly, it could be that—although considering only those three criteria—the transplant surgeon and attending anesthesiologist inadvertently chose recipients of organs with higher fat content or poorer quality to be treated with aprotinin. However, a sensitivity analysis controlling especially for these potential confounders confirmed the independent association of aprotinin with the increased risk of EAD.

Aprotinin is a protease inhibitor derived from bovine or porcine lungs. As such, aprotinin inhibits human proteases such as, for example, trypsin or kallikrein but moreover plasmin, thus decelerating hyperfibrinolysis. In 1989, aprotinin was first demonstrated to reduce blood loss, transfusion requirements and duration of surgery in liver graft recipients [14], a finding that was later confirmed in single and multicenter studies [10,30,31,32]. We could not demonstrate any effect of aprotinin on hyperfibrinolysis in our sample. This may be attributed to the small sample size of the matched cohort in combination with the overall low hyperfibrinolysis incidence of 8.3% but also due to our standardized transfusion regimen [7,20]. Moreover, evidence questioning the usefulness of aprotinin accumulated since large multicenter trials had found an increased risk of major cardiac events, stroke and mortality in cardiac-surgery patients [33,34]. This ultimately led to the withdrawal of aprotinin in many countries. In contrast, a study by Schofield and colleagues demonstrated that the incidence of hyperfibrinolysis was higher in patients who had not received aprotinin, but these patients did not require more transfusions [35].

Aprotinin is a serine protease inhibitor. Of these proteases, interaction with plasmin and kallikrein are likely the most important, for hemostasis and the reduction of inflammation. Additionally, aprotinin also inhibits matrix metalloproteinases (MMP). MMPs are zinc-binding proteolytic enzymes, which are responsible for the degradation of extracellular matrix proteins and basement membranes. Detriments in transplant livers following cold storage and ischemia reperfusion are essentially the liver sinusoidal endothelial cells (LSEC) [36]; for example, an eight-fold increase of MMP-9-levels has been shown 30 min after reperfusion in human OLTs [15]. The inhibition of MMP—especially MMP-9—in liver injury has a relevant impact on the attenuation and repair of LSEC [36]. An intravenous infusion of aprotinin during the liver transplantation leads to an unselected inhibition of MMPs, hereby inhibiting bone marrow progenitor cells, which would be needed to repair injured LSEC. Wang et al. described an even more pronounced effect when the MMP-2,9 inhibitor is directly injected to the donor organ in a steatotic rat liver I/R model [36]. As the addition of aprotinin to the organ preservation solution has already been demonstrated to decrease reperfusion injury in lung transplantations [37], this may be a future approach to attenuating the injury of marginal liver grafts.

Aprotinin is transiently stored in the tubular cells for 5–6 days [38]. This may explain the significant difference of renal impairment in aprotinin treated patients within one week after the transplantation. However, the 30 day as well as the 1-year survival in the patients treated with aprotinin did not differ, although the protective effect of aprotinin on renal function in liver transplant patients has been described [11]. We cannot rule out a bias in patient selection for aprotinin treatment due to the treating anesthesiologists’ experience in transplantation of EDC organs as well the clinical impression of the patient awaiting the liver transplantation. Histological quantification of hepatic steatosis is strongly observer dependent and not always reproducible at the transplant site [21,29,39].

## 5. Conclusions

The idea of pharmacologic preconditioning by aprotinin of the OLT recipient in the case of liver transplantation of ECD organs did not improve the short nor the long-term outcomes in this study. Regarding our results presented in this study, we changed our standard protocol for liver transplantations and do not further applicate aprotinin regularly.

## Figures and Tables

**Figure 1 jcm-10-05232-f001:**
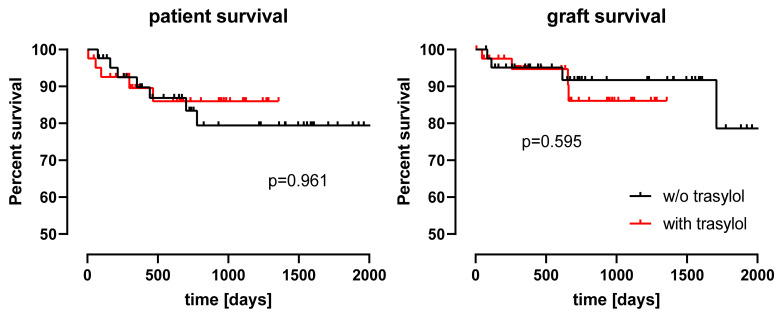
Patient survival and graft survival displayed for treatment with and without (w/o) trasylol. Patient survival (**left**) and graft survival (**right**) did not differ between 42 liver graft recipients receiving aprotinin and 42 matched controls.

**Table 1 jcm-10-05232-t001:** Demographic and clinical characteristics of 42 liver graft recipients were intraoperatively treated with aprotinin and 42 controls. The 42 statistically similar recipients who were not treated with aprotinin were identified by propensity score matching in a cohort of 274 single center liver graft recipients. Propensity scores were calculated using organ quality, cold ischemia time and donor age.

	Whole Cohort			Matched Cohort	
	Controls (*n* = 232)	*p*	Aprotinin (*n* = 42)	*p*	Controls (*n* = 42)
Age (years)	53.5 ± 10.8	0.048	57.0 ± 6.6	0.937	57.1 ± 7.2
Sex (female. *n*)	79 (34.1%)	0.214	10 (23.8%)	0.601	10 (23.8%)
BM (kg/cm^2^)	26.6 ± 5.2	0.702	27.0 ± 5.3	0.361	28.1 ± 5.7
labMELD score	19.9 ± 10.9	0.993	19.9 ± 8.1	0.061	16.4 ± 8.6
Reason for transplantation					
Alcoholic Cirrhosis (n)	57 (24.7%)	0.784	15 (35.7%)	0.596	17 (40.5%)
Hepatocellular Carcinoma (n)	56 (24.2%)		9 (21.4%)		14 (33.3%)
Acute Liver Failure (n)	28 (12.1%)		3 (7.1%)		1 (2.4%)
Primary Biliary Cirrhosis (n)	22 (9.5%)		2 (4.8%)		2 (4.8%)
HBV or HCV Cirrhosis (n)	15 (6.5%)		3 (7.1%)		2 (4.8%)
Graft Failure (n)	14 (6.1%)		3 (7.1%)		0 (0.0%)
Nonalcoholic Steatohepatitis (n)	7 (3.0%)		2 (4.8%)		2 (4.8%)
Other (n)	33 (14.2%)		5 (11.9%)		4 (9.5%)
Preoperative Clinical Chemistry					
Creatinine (mg/dl)	1.6 ± 1.5	0.342	1.8 ± 1.5	0.189	1.4 ± 1.4
AST (U/L)	374.2 ± 1297.0	0.568	518.5 ± 1998.9	0.188	80.6 ± 100.7
ALT (U/L)	303.3 ± 1081.7	0.859	269.5 ± 1007.9	0.205	54.2 ± 55.3
Bilirubin (mg/dL)	7.7 ± 10.0	0.672	7.0 ± 8.5	0.113	4.3 ± 6.3
GGT (U/L)	177.8 ± 239.1	0.694	160.8 ± 278.4	0.817	173.4 ± 184.2
GLDH (U/L)	191.9 ± 936.0	0.682	275.2 ± 1274.6	0.249	8.8 ± 10.6

BMI: body mass index; MELD: laboratory model of end-stage liver disease; HBV: hepatitis B virus; HCV: hepatitis C virus; AST: aspartate transaminase: ALT: alanine transaminase; GGT: gamma glutamyl transpeptidase; GLDH: glutamate dehydrogenase; 4F-PCC: four-factor prothrombin complex concentrate.

**Table 2 jcm-10-05232-t002:** Demographic and clinical characteristics of the liver graft donors and grafts data for 42 liver graft recipients who were intraoperatively treated with aprotinin and 42 controls. The 42 statistically similar recipients who were not treated with aprotinin were identified by propensity score matching in a cohort of 274 single center liver graft recipients. Propensity scores were calculated using organ quality, cold ischemia time and donor age.

	Whole Cohort			Matched Cohort	
	Controls (*n* = 232)	*p*	Aprotinin (*n* = 42)	*p*	Controls (*n* = 42)
Age (years)	54.3 ± 16.7	0.021	59.0 ± 10.8	0.354	61.6 ± 14.5
Sex (female)	107 (46.1%)	0.243	24 (57.1%)	0.127	17 (40.5%)
BMI (kg/cm^2^)	28.8 ± 7.4	0.001	34.7 ± 8.9	0.018	30.3 ± 7.5
Clinical chemistry					
AST (U/L)	108.1 ± 182.6	0.564	127.9 ± 256.8	0.869	72.2 ± 75.9
ALT (U/L)	93.3 ± 223.2	0.837	85.7 ± 114.2	0.032	43.6 ± 38.6
Bilirubin (mmol/L)	0.8 ± 1.1	0.755	0.7 ± 0.9	0.193	1.1 ± 1.5
Sodium (mmol/L)	149.4 ± 10.3	0.984	149.4 ± 7.9	0.182	148.4 ± 7.2
ICU length of stay (days)	4.9 ± 5.1	0.132	6.1 ± 9.7	0.587	6.4 ± 8.0
Organ Quality ^a^					
good	188 (81.0%)	0.001	12 (28.6%)	0.960	12 (28.6%)
acceptable	30 (12.9%)		22 (52.4%)		21 (50%)
poor	14 (6.0%)		8 (19%)		9 (21.4%)
Graft fat content					
Macrovesicular (%)	21.3 ± 22.8	0.057	32.1 ± 23.0	0.319	23.5 ± 25.1
Microvesicular (%)	42.3 ± 28.5	0.198	50.6 ± 22.9	0.246	42.7 ± 29.9
Time to transplantation					
Cold Ischemia Time (min)	496.8 ± 125.3	0.195	524.6 ± 137.5	0.281	509.5 ± 119.0
Warm Ischemia Time (min)	45.0 ± 8.1	0.363	46.2 ± 8.0	0.593	44.2 ± 9.1
Extended donor criteria					
Age > 65 years (n)	62 (26.7%)	0.851	12 (28.6%)	0.340	17 (40.5%)
BMI > 30 (n)	61 (26.3%)	0.001	24 (57.1%)	0.008	12 (28.6%)
ICU stay > 7 days (n)	50 (21.6%)	0.312	7 (16.7%)	0.287	11 (26.2%)
Elevated Transaminases ^b^ (n)	49 (21.1%)	0.416	10 (23.8%)	0.154	5 (11.9%)
CIT > 10 h	41 (17.7%)	0.079	12 (28.6%)	0.450	9 (21.4%)
Bilirubin > 3 mmol/L	9 (3.9%)	0.530	1 (2.4%)	0.645	5 (11.9%)
Steatosis > 40%	45 (19.4%)	0.001	20 (47.6%)	0.268	15 (35.7%)
Sodium > 165 mmol/L	19 (8.2%)	0.345	2 (4.8%)	0.645	3 (7.1%)
Number of Extented donor criteria					
0 (n)	64 (27.6%)	0.111	4 (9.5%)	0.682	4 (9.5%)
1 or 2 (n)	138 (59.5%)		28 (66.6%)		32 (76.2%)
≥ 3 (n)	29 (12.5%)		10 (23.8%)		6 (14.3%)

^a^ organ quality definition as described by Kork et al. [7]; ^b^ ALT > 105 U/L or AST > 90 U/L per definition of the marginal donor criteria of Eurotransplant [24], BMI: body mass index; AST: aspartate transaminase: ALT: alanine transaminase; ICU: intensive care unit; GGT: gamma glutamyl transpeptidase; GLDH: glutamate dehydrogenase; 4F-PCC: four-factor prothrombin complex concentrate.

**Table 3 jcm-10-05232-t003:** Intraoperative and postoperative complications and mortality of 42 liver graft recipients who were intraoperatively treated with aprotinin and 42 propensity score matched controls. The 42 statistically similar recipients who were not treated with aprotinin were identified by propensity score matching in a cohort of 274 single center liver graft recipients. Propensity scores were calculated using organ quality, cold ischemia time and donor age.

	Whole Cohort			Matched Cohort	
	Controls (*n* = 232)	*p*	Aprotinin (*n* = 42)	*p*	Controls (*n* = 42)
Intraoperative Complications					
Postreperfusion Syndrome (n)	72 (31.0%)	0.013	22 (52.4%)	0.414	20 (47.6%)
Hyperfibrinolysis (n)	10 (4.3%)	0.319	3 (7.1%)	1.000	4 (9.5%)
Intraoperative Transfusions					
Packed Red Blood Cells (U)	8.1 ± 7.5	0.296	13.7 ± 32.0	0.374	8.8 ± 6.9
Fresh Frozen Plasma (U)	15.7 ± 8.3	0.744	15.2 ± 9.0	0.380	17.1 ± 9.5
Platelets (U)	0.9 ± 1.3	0.267	1.2 ± 1.2	0.191	0.8 ± 1.1
Fibrinogen (g)	2.3 ± 2.9	0.013	3.6 ± 3.2	0.094	2.4 ± 3.0
4F-PCC (IU)	1010.9 ± 1534.3	0.492	1200.0 ± 1450.1	0.566	986.8 ± 1710.4
Postoperative Complications					
Early Allograft Dysfunction (n)	72 (31.0%)	0.001	27 (64.3%)	0.029	17 (40.5%)
Rejection Episodes (n)	51 (22.0%)	0.406	12 (14.3%)	0.241	6 (7.1%)
Acute Kidney Injury (n)	51 (22.0%)	0.001	20 (48.8%)	0.033	11 (26.2%)
Renal Replacement Therapy (n)	31 (13.4%)	0.001	10 (24.4%)	0.015	3 (7.1%)
Retransplantation					
30 Day Retransplantation (n)	7 (3.0%)	0.070	4 (9.5%)	0.167	1 (2.4%)
1 Year Retransplantation (n)	10 (4.3%)	0.242	4 (9.5%)	0.693	3 (7.1%)
Reasons for Retransplantation					
Arterial Thrombosis (n)	0 (0%)	0.036	1 (2.4%)	0.306	0
Primary Non-Function (n)	6 (2.6%)		3 (7.1%)		1 (2.4%)
Ischemic Type Biliary Lesions (n)	2 (0.9%)		0 (0.0%)		2 (4.8%)
Tumor (n)	1 (0.4%)		0 (0.0%)		1 (2.4%)
Postoperative Clinical Chemistry					
Creatinine. peak (mg/dL)	2.3 ± 1.7	0.150	2.7 ± 1.6	0.812	2.6 ± 2.2
AST. peak (U/L)	1743.9 ± 2317.2	0.001	3193.6 ± 2273.5	0.010	1984.2 ± 1883.9
ALT. peak (U/L)	1024 ± 1212.6	0.004	1611.3 ± 1152.4	0.007	975.2 ± 937.4
Bilirubine. peak (mg/L)	6.3 ± 5.5	0.155	7.6 ± 5.1	0.167	6.2 ± 4.2
GGT. peak (U/L)	436.4 ± 349.7	0.413	502.9 ± 499.7	0.405	424.1 ± 349.1
GLDH. peak (U/L)	731.6 ± 1285.4	0.025	1199.4 ± 894.8	0.459	1002.3 ± 1464.7
Mortality					
30 Day Mortality (n)	8 (3.4%)	0.654	2 (4.8%)	0.152	0 (0.0%)
1 Year Mortality (n)	22 (9.5%)	0.580	5 (11.9%)	0.724	4 (9.5%)
Overall Mortality (n)	31 (13.4%)	0.810	6 (14.3%)	0.763	7 (16.7%)

4F-PCC: four-factor prothrombin complex concentrate; AST: aspartate transaminase: ALT: alanine transaminase; ICU: intensive care unit; GGT: gamma glutamyl transpeptidase; GLDH: glutamate dehydrogenase.

**Table 4 jcm-10-05232-t004:** Intraoperative treatment with aprotinin is associated with early allograft dysfunction (EAD) and peak aspartate transaminase (AST) after liver graft transplantation. Two multivariable models in a cohort of 84 liver graft recipients (42 treated with aprotinin and 42 propensity score matched controls) describe this association. On the left, a binary logistic regression model for EAD and on the right a linear regression model for peak AST.

	Early Allograft Dysfunction			Peak AST after Transplantation		
	OR	(95%CI)	*p*	beta	(95%CI)	*p*
Recipient data						
Age (years)	0.95	(0.86–1.04)	0.242	−33.4	(−107.8–41.0)	0.373
Sex (female)	0.64	(0.15–2.66)	0.534	−437. 6	(−1632.9–757.7)	0.467
BMI (kg/cm^2^)	1.02	(0.9–1.15)	0.768	−19.0	(−120.3–82.3)	0.709
MELD score	0.98	(0.92–1.06)	0.671	−47.4	(−104.7–10.0)	0.104
Donor data						
Age (years)	0.98	(0.94–1.03)	0.478	−64.0	(−103.7–−24.3)	0.002
BMI (kg/cm^2^)	0.96	(0.89–1.02)	0.181	−22.9	(−79.6–33.9)	0.423
AST (U/L)	1.00	(1.00–1.00)	0.833	−2.0	(−4.6–0.5)	0.115
Cold Ischemia Time (minutes)	1.00	(1.00–1.01)	0.994	1.4	(−2.5–5.2)	0.483
Organ Quality ^a^						
acceptable	4.95	(1.26–19.46)	0.022	1770.0	(814.4–3031.2)	0.001
poor	11.78	(1.99–69.55)	0.007	1324.8	(387.6–3152.4)	0.013
Treatment with Aprotinin	4.12	(1.21–14.00)	0.023	−33.4	(354.0–2295.6)	0.008

^a^ compared to good organ quality, definition as described Kork et al. [7] BMI: body mass index; MELD: laboratory model of end-stage liver disease; AST: aspartate transaminase.

## Data Availability

The datasets generated and/or analyzed during the current study are not publicly available due to German Data Protection laws but are available from the corresponding author on reasonable request after approval of the local ethics committee and data safety board.

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
