# Peer review of "Effect of Aprotinin on Liver Injury after Transplantation of Extended Criteria Donor Grafts in Humans: A Retrospective Propensity Score Matched Cohort Analysis"

_jcm, 2021, doi:10.3390/jcm10225232_

Round 1
Reviewer 1 Report
This study is a comparative, retrospective study which analyzed the outcomes after aprotinin for extended criteria of liver graft.
Comments
- The rationale is not clear. As the authors described in Discussion section (which may be more appropriate in Introduction section) in line 267-270, why the authors restarted this abandoned drug for analysis and did not need IRB approval?
- The Introduction section is not completed. Line 35 seems truncated and line 37-40 seems a part of Result section. Besides, introduction of aprotinin and the rationale to use it in this study is missing.
- Rationale of design is confusing. Line 128-132, you used three criteria to decide whether aprotinin is used in the patient by expert opinions, and at the same time, you used these three criteria to perform propensity matching to select cases for control. So there must be some patients who meet the criteria to receive aprotinin treatment but actually not. What is the actual SOP?
- Data presentation is not completed. The authors should list the whole cohort (n = 290), along with the matched cohort, analyzed and presented in Table 1-3.
- The matching strategy should be declared in Method section. Besides, are the results still consistent using backward selection process? Do you describe linear regression model (in Table 4) methodology in Method section?
- Grammar error in Conclusion, Line 318-319.
Author Response
Reply to Reviewer 1
This study is a comparative, retrospective study which analyzed the outcomes after aprotinin for extended criteria of liver graft.
Comments
- The rationale is not clear. As the authors described in Discussion section (which may be more appropriate in Introduction section) in line 267-270, why the authors restarted this abandoned drug for analysis and did not need IRB approval?
We apologize for the inconvenience, but the introduction was truncated in the submission process. The introduction is now complete (line 42-66), and it states the rationale why aprotinin was used in liver transplant recipients at our center as well as the rationale for this retrospective analysis. For further clarification: At that time, we treated liver graft recipients with aprotinin to reduce the reperfusion injury based on evidence from the literature and not to reduce blood loss. The application of aprotinin has to be considered off-label use but off-label use does not require IRB approval in Germany.
- The Introduction section is not completed. Line 35 seems truncated and line 37-40 seems a part of Result section. Besides, introduction of aprotinin and the rationale to use it in this study is missing.
We apologize for the mistake. The introduction section was unfortunately truncated. It is now complete (please also see answer to question 1). The missing part of the introduction sheds light on the mechanisms how aprotinin could reduce blood loss, how aprotinin could leads to better graft survival, why it has been taken off the market, and why it remained interesting for the transplantation of liver grafts of marginal quality (line 47-61).
- Rationale of design is confusing. Line 128-132, you used three criteria to decide whether aprotinin is used in the patient by expert opinions, and at the same time, you used these three criteria to perform propensity matching to select cases for control. So there must be some patients who meet the criteria to receive aprotinin treatment but actually not.
The reviewer is absolutely right. Whether the recipient was treated with aprotinin was a shared clinical decision between the attending surgeon and anesthesiologist, considering the three mentioned factors. This was clarified in the methods section (line 100-103). The decision was neither based on an algorithm, nor was there a cut-off regarding the number of fulfilled criteria. In fact, as the reviewer correctly pointed out, there were 5 patients receiving aprotinin with 0/3 criteria fulfilled, 20 patients with 1/3 criteria, 16 with 2/3 criteria and 1 with 3/3 criteria fulfilled. These numers are also included in Table 2 at the bottom.
- What is the actual SOP?
The current standard operating procedure for liver transplantation does not advise to applicate aprotinin.
- Data presentation is not completed. The authors should list the whole cohort (n = 274), along with the matched cohort, analyzed and presented in Table 1-3.
Upon the request of the reviewer, we included the whole cohort of 274 liver transplantations alongside the matched cohort in all three tables.
- The matching strategy should be declared in Method section. Besides, are the results still consistent using backward selection process? Do you describe linear regression model (in Table 4) methodology in Method section?
The matching strategy, logistic regression and the linear regression methodoly is now described in the methods section (line 139–151).
- Grammar error in Conclusion, Line 318-319.
The mistake has been corrected (now line 559–562).
Reviewer 2 Report
Dear Authors
Congratulations on writing the article “Effect of aprotinin on liver injury after transplantation of extended criteria donor grafts in humans: A retrospective propensity score matched cohort analysis ”. It is a retrospective study analysing the authors experience with the use of aprotinin in liver transplantation. As all retrospective studies, it carries the hurdle of bias and the weakness of its outcomes. In order to overcome this issue and compare the effects of this drugs on surgical outcomes, a propensity score match analysis was performed for donor, graft and patient variables. After matching the groups, 42 patients receiving aprotinin and 42 controls patients (no aprotinin) were selected and compared.
Overall, the article seems structured and the statistical analyses seems fair however some structural and some points will need revision before major decisions could be made.
Please find bellow some points that I have raised:
- Introduction: it is a copy of the Abstract. It needs to be completely re-written.
- Donor data: most studies using extended donor criteria and marginal livers offer the donor DRI, a well stablished metric in donor quality. The authors should state why they did not use it.
- Page 2 line 60 – reference format (upper case brackets) is different from others references
- Page 2 line 61- Eurotransplant criteria – needs reference
- labMELD – should be changed to MELD in the text and in the tables. It should also be clear that it is not NaMELD or MELD3.0
- Page 2 – Aprotinin application: here comes the selection bias of most retrospective studies. Values and cut offs (even if not precisely) should be stated. The three variables (visual assessment, CIT and donor age) were selected based on: visual assessment of fat done by the surgeon at the back table or during implantation? The same surgeon doing the transplant decides at the start of the transplant or during the transplant? Old school anaesthesiologists would give aprotinin for cirrhotic patients due to alcoholic disease because they believe that they tend to bleed more than non alcoholic patients. CIT – what was roughly the cut off value? And donor age? Moreover, how many surgeons performed the transplants in these 5 year’s time?
- Standard definition of Acute Kidney failure should be stated and referred.
- Definition of hyperfribrinolysis should be made and referred.
- Page 3 line 119 and others, including the graphic page 7: trasylol should be changed for aprotinin
- Results – page 3 line 128 to 131: should be in methods, in the aprotinin application section
- Page 5 line 179: wording is confusing and needs to be re-structured, for the sake of clarity.
- Discussion (page 8 and 9)
- Overall addressed most issues of the study however it does not read smoothly, with breaks and even a subsection
- Why a sub-section of discussion named “renal impairment”? This was not the focus of the study.
- Line 230 – missing ‘were”
- Aprotinin dose and timing should be raised as a concern
- Only one case of HAT (hepatic artery thrombosis) in 42 patients (2.3%). Is this a perioperative HAT or a HAT seen later in the transplant?
- My last question: the aim of the study was not to assess aprotinin as a liver preconditioning drug but to assess the consequences (advantages/disadvantages) of using aprotinin in the liver transplantation. As it is a retrospective study (with many bias) the reader will assume that the latter is correct.
Best wishes
Author Response
- Introduction: it is a copy of the Abstract. It needs to be completely re-written.
We apologize for the inconvinience, the introduction was truncated in the submission process. It is now complete (line 42–66).
- Donor data: most studies using extended donor criteria and marginal livers offer the donor DRI, a well stablished metric in donor quality. The authors should state why they did not use it.
Thank you for the valuable comment. We did not use Donor Risk Index because it is still debated upon (PMID 28590542) Eurotransplant has developed an own DRI (etDRI) which has been established in 2017. As our patient sample is constituted by patients we treated between 2010 and 2015, there was no etDRI for those patients at the time. However, our analyses include most of the variables included in the DRI. We statistically accounted for DRI by the variables in the models and the propensity score matching.
- Page 2 line 60 – reference format (upper case brackets) is different from others references
Reference format has been adapted throughout the text
- Page 2 line 61- Eurotransplant criteria – needs reference
The reference has been included (page 2, line 35).
- labMELD – should be changed to MELD in the text and in the tables. It should also be clear that it is not NaMELD or MELD3.0
The requested changes have been made to the manuscript. labMELD has been replaced by MELD in the whole manuscript, except in the definition section in the methods section clearly states that the labMELD was calculated and neither the NaMELD nor the MELD3.0.
- Page 2 – Aprotinin application: here comes the selection bias of most retrospective studies. Values and cut offs (even if not precisely) should be stated. The three variables (visual assessment, CIT and donor age) were selected based on: visual assessment of fat done by the surgeon at the back table or during implantation? The same surgeon doing the transplant decides at the start of the transplant or during the transplant? Old school anaesthesiologists would give aprotinin for cirrhotic patients due to alcoholic disease because they believe that they tend to bleed more than non alcoholic patients. CIT – what was roughly the cut off value? And donor age? Moreover, how many surgeons performed the transplants in these 5 year’s time?
The reviewer is absolutely right. Whether the recipient was treated with aprotinin was a shared clinical decision between the attending surgeon and anesthesiologist, considering the three mentioned factors. This was clarified in the methods section (line 100-103). The decision was neither based on an algorithm, nor was there a cut-off regarding the number of fulfilled criteria. They were – as correctly suspected by the reviewer – visual assessment at the back table before transplantation by the transplanting surgeon. At that time 4 surgeons performed liver transplantations. For further clarification, the count of fulfilled criteria are depicted in Table 2 as well as the values for the three criteria are included in all three tables for both the matched und the unmatched cohort.
- Standard definition of Acute Kidney failure should be stated and referred.
The use of the AKI definition by the KDIGO clinical practice guideline is now referenced in the methods section of the manuscript.
- Definition of hyperfribrinolysis should be made and referred.
Thank you for pointing this out. Hyperfibrinolysis was diagnosed by intraoperatively conducted rotational thrombelastometry and diagnosed according to the Essener Runde algorithm. Respective changes were made to the manuscript including the appropriate reference.
- Page 3 line 119 and others, including the graphic page 7: trasylol should be changed for aprotinin
Again, thank you for pointing this out, the requested change has been made; the whole manuscript was revised and trasylol has been changed to aprotinin.
- Results – page 3 line 128 to 131: should be in methods, in the aprotinin application section
The respective text passage has been incorporated in the methods section (aprotinin application, line 100-104). However, we left it also in the results for better readability and understanding. This can still be removed, if desired by the reviewer.
- Page 5 line 179: wording is confusing and needs to be re-structured, for the sake of clarity.
The passage has been rewritten and the sentences shortened for better readability and more clarity (page 6, line 209-213).
- Discussion (page 8 and 9)
- Overall addressed most issues of the study however it does not read smoothly, with breaks and even a subsection
- Why a sub-section of discussion named “renal impairment”? This was not the focus of the study.
- Line 230 – missing ‘were”
- Aprotinin dose and timing should be raised as a concern
The remarks 1-4 have been addressed. The whole discussion has been revised. It should read more smoothly with less breaks. The subsection has been elimited, missing words filled in and wording revised for better and smoother readability.
- Only one case of HAT (hepatic artery thrombosis) in 42 patients (2.3%). Is this a perioperative HAT or a HAT seen later in the transplant?
This was a perioperative HAT.
- My last question: the aim of the study was not to assess aprotinin as a liver preconditioning drug but to assess the consequences (advantages/disadvantages) of using aprotinin in the liver transplantation. As it is a retrospective study (with many bias) the reader will assume that the latter is correct.
The Reviewer is absolutely right. As the introduction had been truncated during the submission process, the rationale as to why aprotinin was used at that time as well as why we conducted this retrospective analysis were not clearly stated in the manuscript. With the completion of the introduction, this is now clearly stated at the beginning of the manuscript. Because several questions from Reviewer #1 as well as Reveiwer #2 raised the mentioned concerns, several changes (including integration of the whole cohort) that facilitate bias assessment by the reader have been made during our revision of the manuscript. We thank both Reviewers for their comments that have led to a great improvement of our manuscript.
Round 2
Reviewer 1 Report
I have no other comments.